# Long-Acting Injectable Antipsychotic Treatment during Pregnancy: A Case Series

**DOI:** 10.3390/ijerph20043080

**Published:** 2023-02-09

**Authors:** Georgios Eleftheriou, Raffaella Butera, Anna Sangiovanni, Claudia Palumbo, Emi Bondi

**Affiliations:** 1Poison Control Center and Teratology Information Center, ASST Papa Giovanni XXIII, 24100 Bergamo, Italy; 2Poison Control Center, ASST Papa Giovanni XXIII, 24100 Bergamo, Italy; 3Department of Psychiatry, ASST Papa Giovanni XXIII, 24100 Bergamo, Italy

**Keywords:** long-acting injectable antipsychotics, pregnancy, neonatal outcome

## Abstract

Introduction: Data from the literature show that prolonged-release injectable antipsychotics (LAIs) ensure constant blood drug levels better patient compliance and offer a simpler treatment regimen for both patients and caregivers. This observational–descriptive study aims to detect the possible complications found in newborns of women with bipolar or psychotic disorders and LAI therapy during pregnancy. Methods: This study involved women with psychotic disorders during pregnancy who contacted the Teratology Information Center of Bergamo, Italy between 2016 and 2021 to receive counseling on the possible risks of exposure to LAI therapy. The follow-up procedure was carried out by telephone interview or direct contact with the patient and/or her physician. Results: In this study, LAI treatment in pregnancy was not associated with an increased risk of malformations. All but one of the children in the sample were born healthy and the mothers maintained psychopathological compensation during pregnancy. Conclusions: This study showed that, despite the small size of the sample under examination, the administration of LAIs do not compromise the normal intrauterine development of the unborn child and there were no evident major malformations.

## 1. Introduction

Long-acting injectable antipsychotics (LAIs) such as long-acting aripiprazole and long-acting paliperidone have been formulated and widely utilized for the last decade in order to enhance compliance and thereby outcomes for patients with schizophrenia and bipolar affective disorder. Research data underline the importance of the use of LAI in preventing hospitalization and the relapse of serious mental illnesses as they are considered by the patient to be more manageable and consequently induce greater therapeutic adherence. The good adherence is important during pregnancy because relapse of illnesses in schizophrenia and bipolar disorder increases the risk of poor outcomes (both mother and fetus). In spite of the large experience with the use of the same molecules by oral route without adverse effects on the newborn [1], there is still some concern on the use of these LAIs in pregnancy. Information is limited to few case studies and case series: inconsistent outcomes have been documented, including some negative outcomes (prematurity, babies being small for gestational age, congenital anomalies and neonatal adjustment problems), which are difficult to generalize [2,3,4,5].

A potential advantage of LAIs is due to the fact that LAIs bypass the first-pass metabolism and are associated with more stable plasma drug levels. 

LAIs present different pharmacokinetic proprieties compared with the oral antipsychotic drugs. Long-acting paliperidone and long-acting aripiprazole are injected intramuscularly, and the drug release begins slowly. 

The median time to peak plasma concentration of long-acting aripiprazole occurs within 4 days, following multiple deltoid injections, and 5–7 days, following multiple gluteal injections, and the apparent half-life varies from 29.9 to 46.5 days, while for oral aripiprazole, the time to reach maximal plasma concentration is 3 to 5 h, and the half-life is about 75 h [6].

The time to peak plasma concentration of long-acting paliperidone occurs at a median of 13 days after single injection and its half-life is about 25–49 days, while for oral paliperidone, the time to reach maximal plasma concentration is about 24 h and the half-life elimination is about 23 h [7].

This leads to greater psychiatric stability during pregnancy and may reduce fetal exposure to fluctuating high plasma levels. 

The aim of this study was to describe the outcome after LAI exposure during pregnancy in a case series with detailed information on pregnancy outcome of each patient.

## 2. Detailed Case Description

Pregnant women who contacted the Teratology Information Center—Poison Control Center of Bergamo, directly or through their physician, about gestational exposure to LAIs between 2016 and 2021 were prospectively enrolled in this case series. Our center advises women or their physicians regarding the safety or the risks associated with drug exposures during pregnancy and lactation. Data are routinely collected using a structured questionnaire, which was used for this case series as well. 

At first request and before the pregnancy outcome is known, the following information was collected, and the answers were as detailed as possible:-Sensible data of the patient (name, surname, age, city, province of residence and telephone number) were collected for clinical purposes and handled according to the current standards and laws on data protection;-Drug history, including drug dosage, how long the treatment was necessary and at which week of pregnancy the drug was used or should be used, as well as the week the therapy ended (if ended);-Patient history, including the reason and the disease triggering the prescription; in selected cases and in rare disease clinical reports and files (as needed for the clinical management of the patient) were assessed too; moreover, concomitant medicines, timing and dosage of folic acid consumption as well as if the patient was exposed to illicit drugs, smoking and/or alcohol;-Details about actual pregnancy, including week of pregnancy at the time of first contact, last menstrual cycle, whether pregnancy was unplanned, wanted or using assisted fertilization; during the third trimester of pregnancy, it was checked whether a second-level echography was performed and if it was normal or with pathological evidence;-Details about previous pregnancies, and if any, number of live births and pregnancy losses (spontaneous abortions, intrauterine death, elective termination of pregnancy).

The patient’s consent is provided before any consultation. It was mandatory for any request, and was provided orally, as it was the only possible method considering that contact with our service is only made by phone. The telephone calls were registered for medicolegal reasons, maintaining professional secrecy between the healthcare professional and patient. 

After the expected date of delivery, a pregnancy outcome was actively sought. Follow-up was performed by means of telephone interview to collect information on the pregnancy outcome. The following information was collected and added as follow-up in the patient’s record: outcome as live birth or stillbirth; major or minor malformations; perinatal death; preterm delivery; birth weight; Apgar score at 1 min and 5 min after birth. We used the definitions provided by the World Health Organization: preterm delivery when the birth was provided before 37 weeks, while babies born before 28 weeks of pregnancy were considered extremely preterm. The term low birth weight refers to an absolute weight of <2500 g regardless of gestational age. Very low birth weight is accepted as <1500 g and extremely low birth weight is <1000 g. Miscarriage is defined as spontaneous pregnancy loss before or at 20 completed weeks; stillbirth is defined as spontaneous pregnancy loss beyond 20 completed weeks [8]. Neonatal adaptation disorders include central nervous; gastrointestinal; and respiratory system signs (restlessness; rigidity; jitteriness; feeding difficulty; respiratory distress) that result from drug-induced overstimulation of the central nervous system or, alternatively, withdrawal.

A total of 19 women contacted the Teratology Information Center—Poison Control Center of Bergamo between 2016 and 2021 because of LAI exposure during pregnancy. Among them, 15 pregnancy follow-ups were completed, while in four cases, the patients were lost at follow-up. Involved drugs were long-acting aripiprazole in 10 pregnancies (1 follow-up lost), long-acting paliperidone in 8 (2 follow-ups lost) and long-acting olanzapine in 1 (1 follow-up lost). All women were exposed to LAI in the first trimester and continued LAI treatment until delivery. All women took folic acid 0.4 mg once pregnancy was recognized. None of the patients had gestational diabetes nor obstetric complications. The details on each case (specific LAI dose, timing, duration of exposure, indication, co-exposures in first trimester, use of social drugs, tobacco or alcohol, mode of delivery and outcome) are presented in Table 1. Monthly doses of long-acting aripiprazole were 200 mg in two patients and 400 mg in seven women; monthly doses of long-acting paliperidone were 50 mg in two patients and 100 mg in three pregnant women.

A cesarean section was performed in eight cases and four vaginal deliveries were registered. Maternal and infant characteristics and the pregnancy outcomes are summarized in Table 2. There were two miscarriages and one pregnancy ended in a stillbirth in week 26. Twelve pregnancies resulted in 12 live births with no malformations. One of the newborn babies was affected by Down syndrome and he presented fetal hydrops, complicated by septic shock and renal failure; he died 10 days after delivery. None of the neonates presented adaptation disorders after delivery.

Two of the 12 infants were re-evaluated in a longer follow-up, at 15 and 30 months of age, respectively: no adverse effects on development were reported.

## 3. Discussion

In bipolar disorder treatment, noncompliance represents one of the most important reasons for relapse, which can reach up to 40%. Many authors reported that long-acting injectable antipsychotic treatment is associated with a lower mortality rate, as well as a decrease in caregiver burden and increase in patient satisfaction [9,10]. During pregnancy and postpartum, women with bipolar disorder may suffer from increased vulnerability: careful adjustment of the treatment regimen is needed to attenuate the risk of relapse. Long-acting injectable antipsychotic treatment during pregnancy may increase treatment adherence, decrease aggressiveness and improve overall patient engagement [11].

Our case series provides outcome data on LAI-exposed pregnancies, including detailed documentation of exposure, because of the careful assessment of patient past and current medical history. In contrast to studies based on prescription data only, in our series, drug exposure was precisely assessed. In all patients, the pregnancy started when the patients were already on LAI treatment, and in all cases but one, the therapy was continued with certainty throughout the entire pregnancy, thus allowing us to assess the effects of LAI exposure during all the time periods critical for organ development.

In addition, our data provide information on the entire scope of pregnancy outcomes including spontaneous abortions and stillbirth. Eventually, our study is prospective with the known advantages of tracking participant changes and collecting properly exposure and outcome data.

It is clear that the maim limitation of our case series is the limited number of patients and the lack of a control group. The limited number of patients preclude any meaningful statistical analysis within the group (aimed at identifying possible risk factors for the different outcomes observed). In fact, the very small sample size has low statistical power, the analysis could be underpowered, and the study is unlikely to make correct conclusions.

Studies without a statistical analysis and a comparison group, when appropriately used, may offer a larger base of knowledge for medical reasoning, particularly when a question of importance cannot be addressed by other methods because of ethical constraints, or may become the trigger for a subsequent clinical investigation when this is feasible [12].

In this study, one neonate born at 23 weeks of pregnancy died ten days after delivery. The neonate presented fetal hydrops, complicated by septic shock and renal failure; he was affected by Down syndrome, a chromosomic anomaly that rarely presents with fetal hydrops and anemia. This syndrome cannot be considered a drug-induced malformation [13]. 

The percentage of pregnancy loss found in the present case series is relatively high (20%). Nevertheless, among the 11,072 women and 15,950 pregnancies in the Nurses’ Health Study II, 17.3% ended in spontaneous abortion [14]. Other studies reported that women with psychiatric disorders were associated with an increased risk of miscarriage of nearly 17% [15,16], but it is difficult to distinguish if the overall risk of spontaneous abortion was due to illness factors or medication factors. A systematic review and meta-analysis of second-generation antipsychotics was completed by Terrana et al. This included 12 studies with 1782 exposed pregnancies and 1.3 million control subjects, and no significant detrimental associations were found regarding second-generation antipsychotic use and miscarriages or still births [17].

Another issue that emerges from this study is that a caesarian section was performed in 66.6% of the patients (eight patients over 12 deliveries). Studies on oral antipsychotic use in pregnancy showed a higher risk for caesarian section [18]. Sydsjö et al. reported that the burden of psychiatric illnesses was significantly higher in women giving birth by caesarean section on maternal request (10 versus 3.5%, *p* < 0.001), and in another research, women with bipolar disorders treated with lithium plus antidepressant and without other obstetric complications had a 70.5% probability of C-Section [19,20].

There are many reasons for the option of a C-section instead of a vaginal delivery. Pregnant women with psychiatric illness may have high rates of obesity and gestational diabetes, clinical indication to perform a C-section, but in our case series, none of the patients were diabetic and only one presented a body mass index > 30. The most plausible reason of an elective C-section is the psychiatric vulnerability of the patients and the possible tokophobia; the last topic is more often diagnosed in women suffering from psychiatric disorders and in women who have received psychiatric treatment [21,22].

Medium- and long-term data related to the development of the babies are an important issue that may add an important decision element while prescribing long-acting injectable antipsychotics in pregnancy. Unfortunately, this topic requests resources and knowledge that exceed the capabilities of many teratology information services. In our case series, two of the 12 infants were re-evaluated at 15 and 30 months of age and there were no apparent adverse effects on development. However, we acknowledge that the medium- and long-term outcome for the infants should be assessed by a child neuropsychiatrist: this is not possible to perform easily if there is no medical or psychiatric reason and the mothers do not ask for their intervention.

Eventually, from this study, the very high relapse of psychiatric disorder in the post-partum period (41.6%, five patients in 12 deliveries) needing hospitalization was underlined. It is well known that after delivery, there is a sudden drop in estrogen levels that increases a mother’s vulnerability. Therefore, the perinatal period is a period of high risk of instability for mothers, with particular relevance for those with severe mental disorders [23]. Many authors have reported that recurrence rates in women with bipolar disorders during the first 3–6 months postpartum are at least 20% to 30%; the overall postpartum relapse risk is very high, but insufficient information was available to determine a possible “protective” effect of pregnancy and the risk factor during the post-partum period and further research in this field is necessary [24,25]. It is possible that the severity of the illness experienced by the patients could play a significant role in the observed high relapse rate in our study, but with the methodological approach of the follow-up, it was difficult to directly assess the severity of the bipolar disorder.

A woman with bipolar disorder is always to be considered at more risk to herself and to the fetus than a woman treated with antipsychotics. Obstetric complications and adverse neonatal outcomes have been seen to be more serious when a woman is not treated pharmacologically. Moreover, untreated bipolar disorder and schizophrenia may be considered independent risk factors for congenital malformations: accordingly, the most reasonable choice for treating pregnant women with bipolar disorder appears to be maintaining them at the safest minimum dosage [26,27].

## 4. Conclusions

In conclusion, with the limited data available on pregnant women treated with LAI, our study contributes to the existing literature. The results suggest that LAIs can be a valid therapeutic option in pregnant patients with psychosis, in spite of the lack of large epidemiological data. Given the severity of psychiatric illness, it is important for physicians to consider LAI use during pregnancy, when its use is appropriate.

## Figures and Tables

**Table 1 ijerph-20-03080-t001:** Detailed description of the cases.

LAI Monthly Dose, Timing at First Call	Patient Age	Alcohol, Drug, Tobacco Use	Co-Exposures in First Trimester	Delivery Mode, Outcome, and Comments	APGAR	Hospitalization for Psychosis Relapse
Paliperidone 50 mg, 5 ^w^	26	No/no/no	Lorazepam 1 mg	CS ^a^ 39 ^w^, female 3020 g	9/9	Yes, postpartum
Paliperidone 100 mg, 7 ^w^	32	No/no/no		CS 40 ^w^, male 3250 g	9/10	No
Aripiprazole 400 mg, 6 ^w^	38	No/no/no		Interruption of LAI at 23 ^w^ CS 31 ^w^ 1995 g preterm birth, Down syndrome, fetal hydrops complicated by septic shock, massive anuria and death in 10 days	6/8	Yes, postpartum
Aripiprazole 400 mg, 5 ^w^	25	No/no/no		VD ^b^ female 40 ^w^ 3300 g	9/10	Yes, postpartum
Paliperidone 100 mg, 6 ^w^	30	No/no/no		CS male39 ^w^ 3650 g	10/10	Yes, postpartum
Paliperidone 100 mg, 7 ^w^	25	No/no/no		CS female 40 ^w^ 3255 g	9/10	No
Aripiprazole 200 mg, 10 ^w^	31	No/no/no	Haloperidol 30 mg	Interruption LAI at 14 ^w^ AS ^c^ 15 ^w^	-----	No
Paliperidone 50 mg, 5 ^w^	33	No/no/no	Haloperidol 50 mg	CS, Female 39 ^w^ 3100 g	9/10	No
Aripiprazole 200 mg, 7 ^w^	35	No/no/no	Haloperidol 50 mg	AS 9 ^w^	-----	No
Aripiprazole 200 mg, 10 ^w^	34	No/no/no	Carbamazepine 600 mg, stopped at 10 ^w^	CS male, 40 ^w^ 2900 g	9/10	Yes, postpartum
Aripiprazole 400 mg, 5 ^w^	28	No/no/no	Aripiprazole 15 mg	CS, female 40 ^w^ 3140 g	7/9	No
Aripiprazole 400 mg, 5 ^w^	43	No/no/no		VD male 40 ^w^ 2300 g	10/10	No
Aripiprazole 400 mg, 9 ^w^	31	No/no/no		VD male 40 ^w^ 3500 g	8/10	No
Aripiprazole 400 mg, 8 ^w^	20	No/no/no	Haloperidol 50 mg	Stillbirth 26 ^w^	-----	Yes, after stillbirth
Aripiprazole 400 mg, 8 ^w^	31	No/no/no	Paroxetine 20 mg	VD female 38 ^w^ 3120 g	9/10	No

^a^ CS: cesarian section. ^b^ VD: vaginal delivery. ^c^ AS: spontaneous abortion (miscarriage). ^w^: week of gestation.

**Table 2 ijerph-20-03080-t002:** Pregnancy outcome, infant and maternal characteristics.

	Number	Range
Total cases	19	
Lost FU	4	
Age (median)	31 years	(20–43)
Outcome (*n* = 15)		
Miscarriage	2	
Elective termination	None	
Stillbirth	1	
Malformations	None	
Mode of delivery		
Vaginal delivery	4	
Cesarean section	8	
Delivery resulting in live birth (*n* = 12)	12	
Preterm delivery (<37 gestational weeks)	1	
Term	11	
Birth weight (median)	3044 g	(1995–3650)
Gestational age at birth (median)	38 weeks	(23–40)
Apgar score (median)	8–9	(6–10)
Obstetric complications in pregnancy (*n* = 12 women)	None	
Psychiatric complications in pregnancy (*n* = 12 women)	None	
Psychiatric complications after delivery or miscarriage (*n* = 15 women)	6	

## Data Availability

Data is unavailable due to privacy restrictions.

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
