# Peer review of "Long-Acting Injectable Antipsychotic Treatment during Pregnancy: A Case Series"

_ijerph, 2023, doi:10.3390/ijerph20043080_

Round 1

Reviewer 1 Report

The paper “Long-acting injectable antipsychotic treatment during pregnancy: A case series” demonstrated the effects of antipsychotic drug on the condition of pregnant women with psychotic disorders. This manuscript contains very useful data. However, "Long-acting injectable antipsychotic treatment" in the title is not mentioned enough in the discussion, and a discrepancy between the title and the discussion is observed. This issue is needed to be corrected. All concerns listed below, and the discussion should be rewritten to improve the overall quality.

The reasons for the high incidence of cesarean sections in this disease should be explained.

Medium- and long-term data related to the development of the babies should be needed. It is necessary to prove that the drugs do not affect the children's mental development, etc.

Author Response

We thank the reviewer 1 for his/her suggestions. We changed the discussion and we added the reason of the need of the treatment with LAI in pregnancy mentioning the title of our work. We have discussed the topic of the high incidence of cesarean section observed.

The sentence for C-section is: There are many reasons for the option of a C-section instead of a vaginal delivery. Pregnant women with psychiatric illness may have high rates of obesity and gestational diabetes, clinical indication to perform a C-section, but in our case-series none of the patients was diabetic and only one presented a body mass index > 30. The most plausible reason of an elective C-section is the psychiatric vulnerability of the patients and the possible tokophobia; the last topic is more often diagnosed in women suffering from psychiatric disorders and in women who received psychiatric treatment [21-22].

Unfortunately, it was not possible to have more details about the topic of a medium- and long data related to the development of the babies. We added the sentence about that and the few data of the follow-up.

The new sentence is: Medium- and long-term data related to the development of the babies is an im-portant issue that may add an important decision element while prescribing long-acting injectable antipsychotics in pregnancy. Unfortunately, this topic requests resources and knowledge that exceed the capabilities of many Teratology Information Services. In our case-series, two of the 12 infants were re-evaluated at 15 and 30 months of age and there were no apparent adverse effects on development. However, we acknowledge that the medium- and long-term outcome for the infants should be assessed by a child neuropsychiatrist: this is not possible to do easily, if there is no medical or psychiatric reason and the mothers do not ask their intervention.

Reviewer 2 Report

Dear Authors,

Thnk you for allowing me to review you manuscript. My comments are the following ones.

Please provide more details in the abstract at the results section.

The introduction is well documented.
In the methodology section I recommend to be added more information about the questionnaire, ethical approval and informed consent.
The result section may be enriched with some correlations, especially with questionnaire answers.
The discussion and conclusions are well realized.

Author Response

We thank the reviewer 2 for his/her comments. We added more information about the questionnaire, ethical approval and informed consent and we enriched the results. It was not possible to add statistical correlations with the questionnaire answers because of the very small sample size.

The new sentences for the questionnaire and consensus are: At the first request and before the pregnancy outcome is known, the following information is collected and the answers are as detailed as possible:

- sensible data of the patient (name, surname, age, city and province of residence, tel-ephone number) are collected for clinical purposes and handled according to the current standards and laws on data protection

- drug history, including drug dosage, for how long the treatment was necessary and at which week of pregnancy the drug was used or it should be use, as well as the week of the end of therapy (if ended)

- patient history, including the reason and the disease triggering the prescription; in selected cases and in rare diseases clinical reports and files (as needed for the clinical management of the patient) are assessed too; moreover, concomitant medicines, timing and dosage of folic acid consumption as well as if the patient is exposed to drugs of abuse, smoking and/or alcohol

- details about actual pregnancy, including week of pregnancy at the time of the first contact, last menstrual cycle, if pregnancy was unplanned, wanted or by assisted fer-tilization; during the third trimester of pregnancy, it is checked if a second level echography was performed and if it was normal or with pathological evidences

- details about previous pregnancies, if any, number of live births and pregnancy loss (spontaneous abortions, intrauterine death, elective termination of pregnancy).

Patient’s consensus is given before any consultation. It is mandatory for any re-quest and it is given orally as it is the only possible method in consideration that the contact with our service is made by phone. The telephone calls are registered for medico-legal reasons, maintaining the professional secrecy between healthcare pro-fessional and patient.

After the expected date of delivery, the pregnancy outcome is actively sought. Follow-up is performed by a telephone interview to collect information on the preg-nancy outcome. The following information is collected and added as follow-up in the patient’s record: outcome as live births and stillbirths; major or minor malformations; perinatal death; preterm delivery;

Reviewer 3 Report

This manuscript just descibed some clinical observations. These information might be quite important for clinical pratice. It is unlike a scientific paper which need some real research methods such as statistical analysis. 

Author Response

We thank the reviewer 3 for his/her comment. We agree that describing the clinical data without a statistical analysis is a limitation; from the other hand, the very small sample size haslow statistical power, the analysis could be underpowered and the study is unlikely to make the correct conclusion. Moreover, as Kempen JH reported “when appropriately used they serve an important and legitimate purpose in furthering medical knowledge, particularly when a question of importance cannot be addressed by other methods because of ethical or logistical constraints or as a first step in clinical investigation”. We have added the sentences in our discussion when we describe the limitations of the study.

The new sentence is: It is clear that the maim limitation of our case-series are the limited number of patients and the lack of a control group. The limited number of patients preclude any meaningful statistical analysis within the group (aimed at identifying possible risk factors for the different outcomes observed). In fact, the very small sample size has low statistical power, the analysis could be underpowered and the study is unlikely to make correct conclusions.

Studies without a statistical analysis and a comparison group, when appropriately used, may offer a larger base of knowledge for medical reasoning, in particular when a question of importance cannot be addressed by other methods because of ethical con-straints, or may become the trigger for a subsequent clinical investigation when this is feasible [12].

Round 2

Reviewer 1 Report

The authors replied on each comment sincerely and the replies were appropriate. The quality of papers submitted for consideration includes enough reader's interest and scientific quality. The given paper satisfies requirements for publication of this journal.